# Energy and GHG Emissions Aspects of the COVID Impact in Greece

**Dimitri Lalas [1], Nikolaos Gakis [1], Sebastian Mirasgedis [2], Elena Georgopoulou [2] , Yannis Sarafidis [2] and Haris Doukas [3,*]**

[1] FACE[3]TS S.A., 1 Agiou Isidorou Str., 11471 Athens, Greece; lalas@facets.gr (D.L.); ngakis@facets.gr (N.G.)
[2] National Observatory of Athens, Lofos Nymfon, Thesseon, 11810 Athens, Greece; seba@noa.gr (S.M.); elenag@noa.gr (E.G.); sara@noa.gr (Y.S.)
[3] Decision Support Systems Laboratory, School of Electrical and Computer Engineering, National Technical University of Athens, Iroon Politechniou 9, 15780 Athens, Greece
[*] Correspondence: h_doukas@epu.ntua.gr; Tel.: +30-2107724729

**Abstract:** The effects of COVID-19 have had devasting effects on both health and economies in 2020. At the same time, the lockdown and the downturn of economic activity resulted in a decrease in energy consumption and an accompanying reduction in greenhouse gas emissions. In this article, a comparison with the temperature adjustment of energy use is presented for the main carriers of electricity, natural gas, and oil products in the residential, tertiary, industry, and transport (road transport, domestic aviation, and navigation) sectors in 2020 against the previous two years in Greece, along with the corresponding emissions. As the comparison covers the entire year, both COVID peaks in the March–April and November–December periods and the corresponding lockdown effects as well as seasonal variations are included. The analysis shows a reduction, adjusted for temperature, of 3528 GWh in electricity and 10,286 GWh in transport, and an increase of 1916 GWh in heating and other final uses for a net 11,898 GWh decrease and a resulting emissions reduction of 3.48 MtCO$_2$eq (1.29 MtCO$_2$eq in electricity, 2.69 MtCO$_2$eq in transport, and an increase of 0.54 MtCO$_2$eq in heating), or 4.1%, from total national emissions in 2019. The effect is, to a considerable extent, the result of drastic tourist activity contraction, which is starkly evident in the electricity consumption in the Aegean islands. The comparison between the two lockdown periods brings out clear differences, with the reduction in the second one being considerably smaller as the population reverted, to a large extent, to pre-COVID behavior, which implies that no permanent gains from the COVID long-term impact toward decarbonization should be expected.

**Keywords:** COVID-19; Greece; electricity; greenhouse gas emissions; lockdowns

## 1. Introduction

Greece, as with most other countries, has been affected deeply by the COVID-19 pandemic. Its arrival in Greece dates from mid-February 2020. However, its potential impact was not acknowledged until the beginning of March as the magnitude of the devastation in Italy became apparent.

The Greek Government, in a pro-active response, decreed a lockdown that went into effect on 14 March and lasted for 45 days until 4 May 2020. This first lockdown included schools, retail stores, restaurants, and strict travel restrictions. The general population, possibly because of fear of an unknown threat for which no defense was known or even glimpsed in the horizon, adhered to the instructions, and, as a result, Greece registered one of the lowest number of infections and deaths in Europe.

This 45-day lockdown had an immediate effect on the economy and also on energy demand. An initial analysis of the impact of this spring lockdown is provided in Lalas et al. [1]. The expected reduction in GDP was estimated then by both the Government and the Bank of Greece to reach ca. 5% on a yearly basis but to spring back by an almost equal

amount in 2021 [2]. Other organizations published decidedly gloomier projections, with the European Commission [3] projecting a 9.7% reduction and the International Monetary Fund (IMF) an even higher one. These more pessimistic estimates were based on the large dependence of the Greek economy on tourism. Tourism contributes ca. EUR 18 billion yearly and over 70% of the foreign income which, with indirect effects, amounts to close to 20% of Greek GDP.

With the advent of spring and the improvement of weather conditions, which made possible an increase in outdoor activities, as well as with an eye on the start of the tourist season, the lockdown was lifted on 4 May 2020.

The major effect on the economy was the collapse of the tourist industry. The travel restrictions and the concerns for infection resulted in decreases in incoming tourists and cancellations that in spring reached almost 100%. Even after the travel restrictions were lifted in the summer months, international arrivals were down by more than 70%. The full extent of the impact on the economy is expected to be equally severe as direct income of the tourist sector as estimated by the Greek Tourist Confederation [4] was down by a similar percentage of 75% (31.35 million arrivals and EUR 18.15 billion in 2019 vs. 7.36 million arrivals and EUR 4.32 billion in 2020, respectively). Hence the very high reduction of GDP by over 10% projected now for 2020 by both the Government and the Bank of Greece as well as the IMF, and with the recovery in 2021 projected to also drop from +5.1% to +4.2%, notwithstanding the estimated State support funds to those employees and businesses affected reaching an estimated EUR 23.9 billion with an additional EUR 7.5 billion budgeted for 2021.

Despite efforts to coordinate travel restrictions within the European Union (EU), their lifting in Greece in the summer resulted in a spread of infections. Greece, eager to limit damage to tourism, did not manage tourist flows successfully, while additionally a part of the population did not comply with protective measures. As a result, the number of cases in Greece, as in most of the European Union (EU) Member States, exploded in September, and with more than 90% of the intensive care units (ICUs) full despite their doubling over the summer period, the Greek Government was driven to imposing a second full lockdown on 7 November 2020, including a strict night curfew. The lockdown was partially lifted on 11 January 2021, at which time primary schools opened, while retail stores were allowed to resume activity on 18 January 2021 under specific rules for the number of customers allowed and with time restrictions.

Lockdowns resulted in changes in work and living habits with concurrent reductions in trade, industrial output, and energy use. This led to a noticeable decrease in greenhouse gas (GHG) emissions worldwide, which has been estimated for $CO_2$ energy emissions at ca. 6.7% in 2020, but with substantial differences between the major emitting countries as seen in Table 1 [5] with China on the light end of the spread at −1.7% and the US on the other end at −12.2%.

**Table 1.** Yearly emissions and year-to-year % changes.

| | 2019 | | 2020 | |
|---|---|---|---|---|
| **BilliontCO$_2$/yr** | **Emissions** | **Growth** | **Emissions** | **Growth** |
| China | 10.2 | 2.20% | 10 | −1.7% |
| USA | 5.3 | −2.60% | 4.7 | −12.2% |
| EU27 | 2.9 | −4.50% | 2.6 | −11.3% |
| India | 2.6 | 1.00% | 2.4 | −9.1% |
| World | 36.4 | 0.10% | 34.1 | −6.7% |

The fossil fuel emissions in the EU in 2020 are estimated to drop by 11.2% to 2.6 Gt $CO_2$ from reductions of 12.2% from oil, 20.2% from coal, and only 3.3% from natural gas, (NG) with an additional drop of 5.3% from cement production.

Of interest is also the evolution of the reduction over the course of 2020 for the largest emitters [6]; the difference in timing between China (February peak) and the rest of the

world (April) is clear; the EU is seen to follow the rest of the world with a large dip in April, and a smaller dip in November with the recovery in between to never return fully.

For Greece, the Global Carbon Project [7] estimates that the decrease in $CO_2$ energy emissions will be in the order of 10% (60 $MtCO_2$ in 2020 compared to 67 $MtCO_2$ in 2019, and 72 $MtCO_2$ in 2018 out of a total GHG emissions in 2019 of 92 $MtCO_2$eq), which is close to the EU-27 estimate of $-11.2\%$. In a previous study [1] completed in June 2020, based on data available until the end of May, a 2–2.5 TWh reduction of electricity demand over the full year was estimated resulting in a 1.5–1.9 $MtCO_2$ reduction in GHG missions. For non-electricity energy use in mainly transport and the other sectors, only qualitative estimates were made.

The first lockdown in the spring of 2020 resulted in clear reduction of energy demand in electricity as documented in a number of studies world-wide [8,9] and also in Greece [1]. In particular, Zou et al. [8] based on activity data estimated an 8.8% decrease in global $CO_2$ emissions in the first semester of 2020; McWilliams and Zachman [9] find that weekly reduction peaks in electricity use during the working hours of 08–18 h in EU reached maximum values of 13% during April 2020 and in Greece 20% compared to the same period in 2019, while Lalas et al. [1] estimated for April of the spring lockdown a 9.8% reduction in electricity use and an impressive 41% reduction in oil use for road transport. This is counterintuitive as one would expect consumption to increase in the residential sector as higher use of electrical equipment for telecommunications/on-line work and schooling, as well as other equipment (e.g., for heating, cooling, ventilation, lighting, cooking) to support a longer stay of the members of households at home would have been expected.

In the tertiary sector, on the other hand, during the lockdown period a significant part of the companies either did not operate at all or operated through teleworking, which resulted in a significant reduction in energy consumption. This is especially the case for enterprises in the tourist sector.

The transport sector is affected directly by the imposition of travel restrictions, especially aviation, and increased working at home resulting in reduction of liquid fossil fuel use.

Finally, in industry, conditions during the lockdown created business opportunities for some (leading to increased production) and losses for most industrial sub-sectors as production declined accompanied by a commensurate energy consumption.

The effect of the anti-COVID measures during the first lockdown was noted also in the concentrations of air pollutants in Greece. Koukouli et al. [10] found a ca. 10% of NOx over Greece due to traffic reduction, while Grivas et al. [11] estimated reductions of 53% reduction in $CO_2$ emissions and 50% of $NO_2$ over the city of Athens for the three-week period in April attributed to a 46% traffic reduction.

As more data have become available, it is of interest to examine this pandemic impact on energy demand over the full year of 2020 that includes the seasonal variations in climatic conditions and activity so as to enable a better comparison with that of previous years but also to juxtapose the differences between the spring and winter lockdowns and try to discern whether any of this reduction, if at all, will carry over in the years to come. This is crucial in the way energy use statistics are utilized as guides in energy policy design and used as metrics, especially in the effort to green the economy and mitigate GHG emissions, and this work aims to contribute in putting this on a more solid base.

To this end, in Section 2 the methodological aspects for the analysis of consumption in the main energy carrier (electricity, NG, and oil) including data sources and adjustments due to temperature variations between years are described. In Section 3, following a brief overview of the energy sector, the impact of the lockdown and accompanying measures to address it regarding the three main energy carriers are presented and analyzed. Similarly, in Section 4, after a brief description of the GHG emissions sectoral contribution and their evolution, the resulting reductions compared to those of previous years are presented and discussed. Finally, in Section 5, a discussion of the results and remarks on their implications

are offered.

## 2. Materials and Methods

The approach for the examination of the effect of the measures is based on a comparison of energy demand and consumption during the full 2020 year with that of the two previous years 2019 and 2018. Comparison with additional previous years was not considered as in the years before 2018, the Greek economy was still under the effects of the obligatory reforms of the Memorandum of Understanding (MoU) for financial assistance to overcome its perilous economic circumstances with its creditors (IMF, European Stability Mechanism, European Commission, European Central Bank) with negative growth rates (see Table 1) with the Gross Value Added (GVA) contributions of most sectors reaching their lowest values in the last 15 years. Greece was released from the strict MoU terms in August 2018, although it is required to maintain a 3.5% primary general Government surplus until 2022 and enhanced supervision terms continue to apply. In addition, 2018 and 2019 are the first years since 2009 with a positive national growth rate of 1.9%, which makes comparison easier.

The energy streams to be examined are electricity, natural gas (NG), and oil products. Of the rest two streams, solid fuels and renewables (RES), solid fuels (4979 ktoe in 2019) are used almost exclusively (>95%) for the production of electricity, and similarly of the 3162 ktoe of RES only 810 ktoe of solid biomass are not in electrical form. This solid biomass is mostly used for space heating, but it represents less than 3.7% of gross inland consumption. As a result, the three streams considered, electricity, NG, and oil products including LPG, represent over 95% of the final energy consumption. A description of the data sources used and the temperature adjustment approach for each stream follows.

### 2.1. Electricity Consumption

Currently, most of the Greek islands are not connected to the mainland grid. The electricity on the islands including Crete is generated locally from a combination of internal combustion and steam generators all fueled by oil, plus Photovoltaics (PV) and wind RES. The islands grids are run by DEDDHE, the mainland distribution system operator who publishes load data monthly. The mainland grid is operated by ADMIE, the Greek Transmission System Operator (TSO) who publishes hourly load data and monthly summaries which also include estimates of grid generated electricity mostly from PV, but also submits data to the European National Electricity Transmission Systems Operators (ENTSO-E). For this analysis, daily load values of the mainland grid have been obtained from ENTSO-E and monthly values for the island grids provided by DEDDHE have been used.

As the electricity demand depends on temperature, a correction needs to be applied to make comparisons meaningful. To this effect, the temperature dependence of the load is derived from multi-annual periods and for specific areas [12,13]. For the mainland grid, such a dependence [14] was updated recently [15] based on data for the 13-year 2006–2018 period, as illustrated in Figure 1, and has been utilized in this analysis.

As 2019 was the last pre-pandemic year, it has been chosen to be the basis for comparison, and the temperature adjustments are applied relative to this year's temperature regime. Thus, the polynomial function given in Figure 1 has been utilized to compute the nominal daily load for each day of all three years, 2018, 2019, and 2020, and a correction has been applied to the actual daily loads of 2018 and 2020 which is the difference between the nominal loads as computed by the polynomial function given in Figure 1 between those years and the same day nominal load of the 2019 year.

As no such temperature-load function is available for Crete and the rest of the non-connected grids of the Aegean islands, no temperature correction has been applied to the loads of the islands in view also of the relatively smaller year-to-year temperature variations there.

The temperature data used are the daily average temperature values from the home station of the Greek Meteorological Service at its headquarters in Athens (GR000016716

Hellinikon) downloaded from the US National Center for Environmental Information (NCEI) website.This station has been chosen because it is representative of the greater Athens area which is home to more than 35% of the population of Greece, and 40% of the population of the Aegean Sea islands including Crete, which are not connected to the national grid, are discounted.

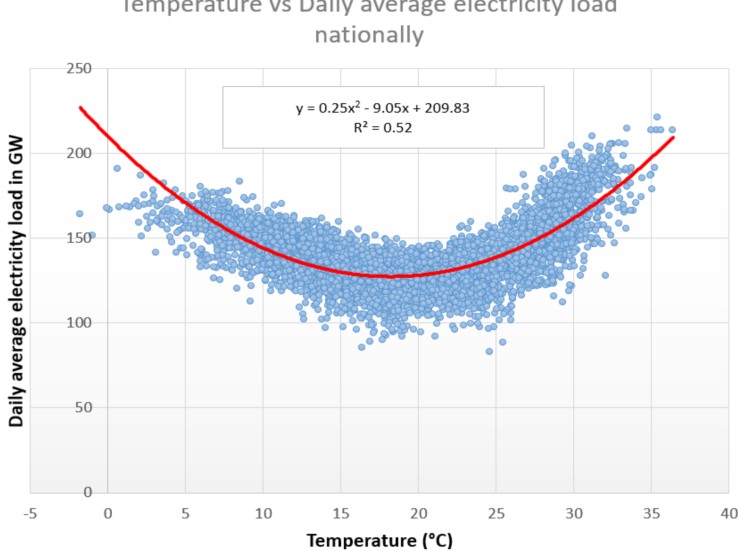

**Figure 1.** Electricity load temperature dependence for the mainland grid of Greece.

### 2.2. Natural Gas Consumption

Data for NG consumption on a monthly basis have been downloaded the daily reports of the Greek NG Transmission System Operator (http://www.xn--mxafd0dp.gr/scada/quantity-hourly.php?la=GR, Last accessed 20 February 2021) as well as from EUROSTAT. The NG consumption covers residential and tertiary, industrial, and power generation needs. As the latter is already included in the electricity demand considerations, the total amount needs to be decomposed accordingly. The dependence of NG consumption on temperature among other factors has been investigated extensively (see [16] for a thorough survey of relevant papers). As for electricity, the temperature dependence is regionally specific, and unfortunately no relation for the NG consumption dependence on temperature is available at this time for Greece. The following procedure for adjustment has been adopted: (a) the monthly amounts for electricity generation have been subtracted from the monthly consumption amounts, (b) then the average monthly amount of consumption during the non-heating months (May to October) has been subtracted from the monthly amounts for the heating months, (c) an adjustment factor for the increase in customers of 0.95 and 1.07 compared to 2019 is applied to the 2018 and 2020 values, (d) an average nominal consumption per Heating Degree Day (HDD) for the winter months of December, January, and February is computed for 2018 and 2020, (e) an adjustment estimate for the retail consumption for heating for 2018 and 2019 heating months is then computed by multiplying this nominal consumption per HDD with the difference in HDDs between 2019 and 2020 or 2018, and finally (f) the total NG consumption for the heating months is derived by adding the value of the heating months for 2018 and 2019 to the adjustment computed in (e). The values for the non-heating months for 2018 and 2020 have been adjusted only for the small annual increase in retail connections of 4% of 2019 compared to 2018 and 5% in 2020 compared to 2019.

The monthly values of HDD used have been obtained from the EUROSTAT data base. Again, the values for the Greater Athens area for the 3-year period of 2018, 2019, and 2020 have been used as proxies for all of Greece. In Figure 2 the variation of HDD as well as Cooling Degree Days (CDD) is presented. The first winter months of 2018 are clearly

warmer than the respective ones in 2019 and 2020, but the last months of 2018 turned cooler with this trend continuing in the winter months of 2019. The cold months of 2019–2020 remained cold, but those of 2017–2018 are seen to be warmer than the same period of 2018–2019, which are similar to those of 2019–2020. The last months of 2020 are clearly warmer than the same ones of 2019.

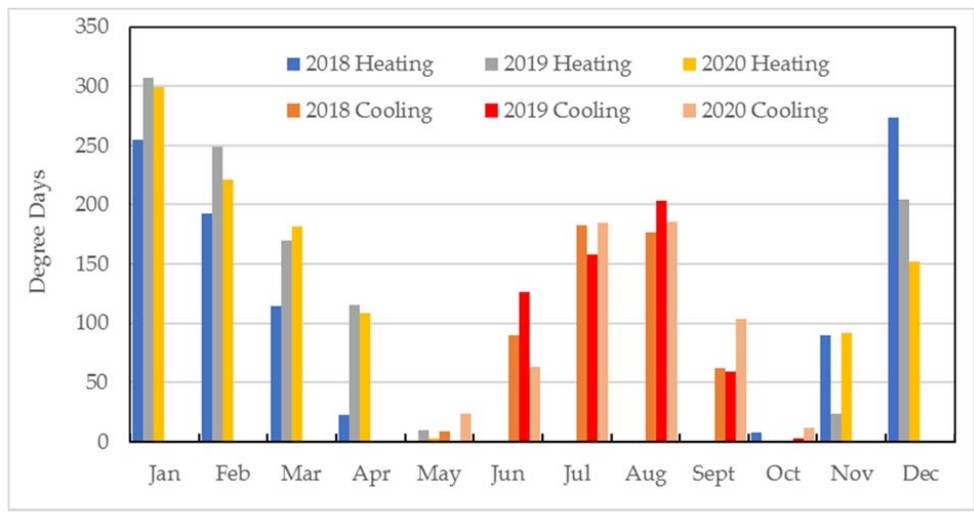

**Figure 2.** Heating (HDD) and cooling (CDD) degree days for the Greater Athens region for 2018, 2019, and 2020.

*2.3. Oil Consumption*

As with NG, the data for oil consumption have been downloaded from the same EUROSTAT energy statistics site. Of the various components of oil consumption, those of Motor Gasoline, Road Diesel, Heating, and other Gas Oil and LPG have been chosen for comparison. LPG is used both for heating/cooking and in road vehicles. The ratio of LPG use in Greece based on the Energy Balance data for both the previous 2018 and 2019 years is ca. 40% and 60% for road, and residential, and industrial use, respectively. The "Heating and other Gas Oil" category according to EUROSTAT definition includes "light heat oil for industrial and commercial use, marine diesel and diesel used in rail traffic and other gas oil used as petrochemical feedstock". From the detailed breakdown of the latest Energy Balances, the amount used in residential, tertiary, and final demand industrial sectors is approximately 75% (78% for 2017, 74% for 2018, and 75% for 2019). As the 2020 Energy Balance is not available, the breakdown of both LPG and Heating and other Gas Oil amounts was carried out using these same 2019 percentages (40% and 75%, respectively) for 2020.

The oil heating amounts for 2018 and 2020 have been adjusted for temperature in the same way as those for NG described above but without any adjustment to account for changes in customer connections. It should be pointed out that heating oil in housing, unlike NG, is sometimes purchased in April when its price is lower because of reduced taxation during the heating period which lasts until 15 April and is stored to be used in the fall. In 2020, this reduced tax period was extended to include May, which would render the heating diesel values for April and possibly May in 2020 less representative of actual consumption.

**3. Impacts on Energy Use**

*Energy Sector Overview*

Gross Inland Consumption in Greece has hovered at ca. 24 Mtoe since 2014 (down from a high of 31 Mtoe in 2009). Greece's only indigenous resources are its good RES potential and lignite [17], used almost exclusively for electricity production, which is scheduled to end completely by end 2023 [18] with only the new Ptolemais V 614 MW plant

to continue operation until the end of 2028. This requires the import of 37,987 ktoe (latest National Energy Balance figures for 2019 as of January 2021), of which 32,245 ktoe are oil and petroleum products. Of that, 18,540 ktoe are exported, with 99% of this comprising oil and petroleum products from the four Greek refineries.

Final energy consumption (FEC) is similarly hovering at ca. 15 Mtoe since 2014 (down from a high of ca. 20 Mtoe in 2008–2009) to a low of ca. 15 Mtoe in 2013–2014 and has remained at that level since. The distribution between sectors (see Table 2) remains in general unchanged over the last 10 years (16.8% for industry, 39.3% for transport, 13.8% for tertiary, 26.7% for households, and about 2% for agriculture in 2019).

**Table 2.** Final energy consumption.

| (ktoe) | 2008 | 2009 | 2010 | 2011 | 2012 | 2013 | 2014 | 2015 | 2016 | 2017 | 2018 | 2019 |
|---|---|---|---|---|---|---|---|---|---|---|---|---|
| **Final energy consumption** | 20,352.2 | 19,656.2 | 18,249.2 | 18,081.9 | 16,278.4 | 14,668.3 | 14,804.1 | 15,741.0 | 15,879.2 | 15,720.8 | 15,168.8 | 15,404.6 |
| Industry sector | 4231.5 | 3462.4 | 3472.8 | 3322.8 | 2982.2 | 2835.5 | 3088.3 | 3128.4 | 3073.1 | 2762.8 | 2743.3 | 2587.6 |
| Iron and steel | 225.2 | 188.4 | 177.1 | 182.7 | 156.2 | 140.8 | 134.7 | 90.7 | 130.7 | 128.5 | 129.6 | 143.5 |
| Chemical and petrochemical | 262.4 | 224.5 | 194.3 | 173.8 | 100.6 | 111.3 | 161.7 | 222.2 | 153.9 | 121.1 | 130.2 | 88.9 |
| Non-ferrous metals | 744.4 | 607.4 | 764.5 | 801.5 | 789.9 | 882.3 | 828.2 | 829.2 | 776.1 | 682.4 | 708.4 | 631.0 |
| Non-metallic minerals | 1132.8 | 856.3 | 968.9 | 726.9 | 684.6 | 727.1 | 760.3 | 736.4 | 776.3 | 680.5 | 649.6 | 641.7 |
| Transport equipment | 34.9 | 34.1 | 25.6 | 37.5 | 17.0 | 12.1 | 20.2 | 21.2 | 15.5 | 18.3 | 22.2 | 7.1 |
| Machinery | 67.3 | 12.2 | 18.9 | 45.8 | 24.4 | 26.5 | 36.4 | 36.6 | 29.6 | 49.9 | 64.8 | 87.2 |
| Mining and quarrying | 91.2 | 76.9 | 59.4 | 33.9 | 64.2 | 74.2 | 75.4 | 87.3 | 80.5 | 90.9 | 115.3 | 89.8 |
| Food, beverages, and tobacco | 658.2 | 618.3 | 580.5 | 595.4 | 539.9 | 470.3 | 522.8 | 523.0 | 445.2 | 423.6 | 459.7 | 455.8 |
| Paper, pulp, and printing | 139.8 | 123.0 | 121.5 | 91.9 | 95.8 | 97.9 | 98.6 | 83.3 | 47.9 | 48.4 | 54.2 | 76.7 |
| Wood and wood products | 51.5 | 43.9 | 48.2 | 54.6 | 37.5 | 29.1 | 24.6 | 30.7 | 23.2 | 27.1 | 41.5 | 29.9 |
| Construction | 152.0 | 150.5 | 128.3 | 87.7 | 52.9 | 86.1 | 151.0 | 127.7 | 128.7 | 165.2 | 152.7 | 134.9 |
| Textile and leather | 169.4 | 93.5 | 88.9 | 76.3 | 46.2 | 43.5 | 32.9 | 31.3 | 41.2 | 38.0 | 98.7 | 86.3 |
| Not elsewhere specified (industry) | 502.4 | 433.3 | 296.7 | 414.9 | 373.2 | 134.4 | 241.5 | 308.9 | 424.3 | 288.8 | 116.3 | 114.7 |
| Transport sector | 7521.5 | 8278.1 | 7352.3 | 6602.1 | 5512.1 | 5608.8 | 5635.5 | 5753.4 | 5897.0 | 5815.3 | 5903.9 | 6046.0 |
| Rail | 44.4 | 38.3 | 24.3 | 20.2 | 30.3 | 27.2 | 57.9 | 59.0 | 56.5 | 55.5 | 54.6 | 26.5 |
| Road | 6505.1 | 7055.9 | 6361.5 | 5833.6 | 4727.6 | 4956.2 | 4925.4 | 4976.1 | 5092.7 | 4992.1 | 5009.0 | 5152.8 |
| Domestic aviation | 352.9 | 290.7 | 237.2 | 221.8 | 187.0 | 176.7 | 179.8 | 167.4 | 189.0 | 195.2 | 222.2 | 224.7 |
| Domestic navigation | 599.1 | 880.7 | 717.1 | 515.6 | 525.2 | 430.5 | 449.1 | 534.3 | 556.8 | 570.6 | 615.7 | 640.7 |
| Commercial and public services | 2230.1 | 2153.7 | 1957.4 | 1870.1 | 1935.3 | 1820.6 | 1714.0 | 1874.9 | 2037.5 | 2191.7 | 2095.3 | 2135.5 |
| Households | 5270.4 | 4887.5 | 4666.5 | 5526.0 | 5096.0 | 3821.3 | 3844.9 | 4460.6 | 4348.7 | 4413.3 | 3916.7 | 4116.2 |
| Agriculture, fishing, and forestry | 1098.7 | 874.6 | 800.2 | 669.1 | 316.0 | 323.9 | 280.6 | 271.3 | 283.4 | 303.6 | 279.2 | 292.2 |
| Not elsewhere specified (other) | 0.0 | 0.0 | 0.0 | 91.7 | 436.7 | 258.2 | 240.9 | 252.3 | 239.4 | 234.0 | 230.5 | 1.3 |

It is of interest, then, to see, to the extent possible, with the currently available energy data, whether and how these FEC trends presented in Table 2 have been affected by the COVID impact.

As the electricity sector is responsible for about one third of all GHG emissions in Greece, we focus on its structure and generation trends first. The total installed capacity on 31 December 2020 was 20,555 MW (see Table 3), of which 9584 MW is conventional and the rest 11,065 MW RES. The total electricity available for consumption is seen in Table 3 to remain virtually unchanged over the last five years with the inland generation variation matched by counterbalancing changes in net imports. However, this pattern

changes substantially in 2020. Consumption in 2020 dropped by 5% and 4% compared to 2019 and 2018, respectively. At the same time, the production pattern also changed notably as lignite production was reduced by 45% compared to 2019 following an almost equal one of 39% between 2019 and 2018 due to the increases in the price of European Trading System (ETS) allowances (EUAs).

In response to the evolution of the COVID-19 cases, the Greek Government has put in place two periods of lockdown, one from 14 March to 4 May and the second from 7 November to the end of the year and beyond into 2021.

In Figure 3, a comparison of the electricity consumption (7-day running averages, after adjustment for temperature as discussed in the previous Section 2) between the pre-COVID 2019 year and 2020 is shown. The consumption for 2018 is also included to illustrate the overall similarity of the yearly pattern of consumption between 2018 and 2019 so as to bring out more clearly the differences.

**Table 3.** Electricity generation, consumption (GWh), and power (MW).

| Year | Lignite | NG | Hydro | Oil | RES | Distribution Grid Gen | Net Imports | Consumption |
|------|---------|-----|-------|-----|-----|----------------------|-------------|-------------|
| 2015 | 19,418 | 7267 | 5391 | 4571 | 6031 | 4714 | 9609 | 57,001 |
| 2016 | 14,898 | 12,512 | 4843 | 4627 | 6519 | 4734 | 8796 | 56,929 |
| 2017 | 16,387 | 15,397 | 3457 | 4927 | 6834 | 4730 | 6237 | 57,969 |
| 2018 | 14,907 | 14,136 | 5051 | 4579 | 7328 | 4732 | 6279 | 57,012 |
| 2019 | 10,418 | 16,228 | 3361 | 4589 | 8150 | 4995 | 9944 | 57,685 |
| 2020 | 5713 | 18,197 | 2819 | 3831 | 9909 | 5580 | 8719 | 54,768 |
| | | | | | | | | |
| Power | Lignite | NG | Hydro | Oil | RES | | | Total |
| Dec 2020 | 2816 | 5011 | 3170 | 1757 | 7895 | | | 20,649 |
| | | | | | | | | |
| RES Power | wind | PV | PV roofs | Small hydro | Bio | Co-gen | | Total |
| Dec 2020 | 4114 | 2833 | 375 | 243 | 96 | 234 | | 7895 |

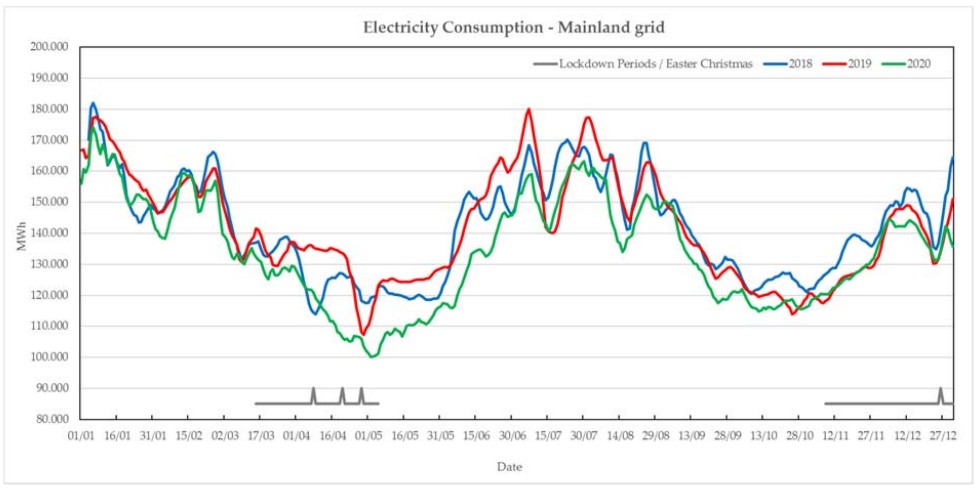

**Figure 3.** Daily 7-day running averages for the electricity consumption in the mainland of Greece for 2019, 2018, and 2020, adjusted for temperature (MWh). The lockdown periods and the Orthodox Easter are also indicated.

The first point to be made is the very high correlation of the yearly patterns (90% to 92% between all three years on a daily basis). In particular, the overall pattern comprised a continuous reduction from the high of the beginning of the year to the end of January;

a double sharp peak in February followed by a second reduction in March; a leveling in April and May; a continuous increase in June as tourism begins reaching a high at the end of the month, when summer leaves typically start and remaining at that level until the beginning of September; a gradual decline then leads to a relative minimum toward the end of October from which point on, again, there is a gradual increase as the days grow shorter until the end of the year. This overall pattern, driven by weather, economic activity, and day length, is almost repeatable for over a decade.

Of note, though, are also shorter sharp variations as can be clearly seen in Figure 3 that can be attributed to specific reasons that do not seem to have been influenced by COVID-19. These include weekends and holidays with those for Easter more noticeable, as it changes dates from year to year (day #98, 8 April for 2018, day #118 28 April for 2019, and day #110 19 April for 2020 for the Orthodox Church). In 2020, the overall minimum is not at Easter as is the case in 2018 and 2019, but after a slight upswing following Easter, almost two weeks later during the long weekend of 1 May at the end of which the lockdown was lifted.

Similar clearly discernable behavior is seen for other important fixed holidays such as 15 August (Dormition of the Virgin Mary) and Christmas. Less important holidays such as Greek Independence Day (day #89, 25 March) and the combined Saint Demetrios–Start of WWII (days #304–306, 26–28 October) may also have some effect, with their magnitude varying from year to year, especially when they result in extended weekends.

The overall difference between 2018 and 2019 is seen to be small. Both the actual (51.21 TWh) and adjusted for temperature (51.83 TWh) consumption for 2018 differ only slightly, i.e., by less than 1% from that of 2019 (51.68 TWh). This is to be compared to the actual (48.91 TWh) and adjusted (48.91 TWh) consumption for 2020, which is more than 5.5% below 2019. It should be noted that these figures refer to the mainland grid, which does not include Crete and the majority of the rest of the Aegean islands.

The overall effect, then, of the COVID-19 measures and consumer behavior in mainland Greece can be seen in Figure 4 and is assessed on a yearly basis at ca. 5% (2.76 TWh) on adjusted consumption. This, however, is not evenly distributed between the two lockdown periods. In the first lockdown period (14 March to 4 May—51 days), the difference between electricity consumption between 2020 and 2019 was 684 GWh, and in the second (7 November to 31 December—54 days) it was only 62 GWh, even though the two periods are of the same duration (51 vs. 54 days until 31 December). This seems to be caused by the difference of general population conduct during the two periods, with that of the first being much more restrained abetted by a reduction of the Government guidelines for teleworking from 80% in the first lockdown as opposed to 50% in the second.

The impact of the pandemic is also noticeable in the consumption of the high voltage customers who represent large industrial installations. The high voltage consumption represents ca. 15% of the total demand and is mostly unaffected by variations of temperature, and consequently provides a good estimate of the lockdown effect on the activity of the industry sector. Whereas, the mean monthly difference between 2018 and 2019 is less than 8%, with the exemption of July in which it reaches a 12.3% peak (see Figure 5); the monthly differences between 2020 and 2019 are much higher, with that for March exceeding 12% and reaching 30% in April, after which time they gradually decrease going below 10% after July and remaining at that lower level (similar to the 2018–2019 one) during the months of the second lockdown in November and December.

This pattern of year-to-year monthly differences is the same with that of the overall demand, but with the decrease in the latter in April and May only reaching 13% and 12%, respectively, i.e., three times smaller than that of the high voltage consumption.

It is of interest to also examine the effect of the pandemic on the islands of the Aegean which are not connected to the mainland grid and where local demand is met by oil fired stations and RES installations. The electricity consumption of Crete and the rest of the Aegean islands s is given separately in Figure 6a,b, respectively.

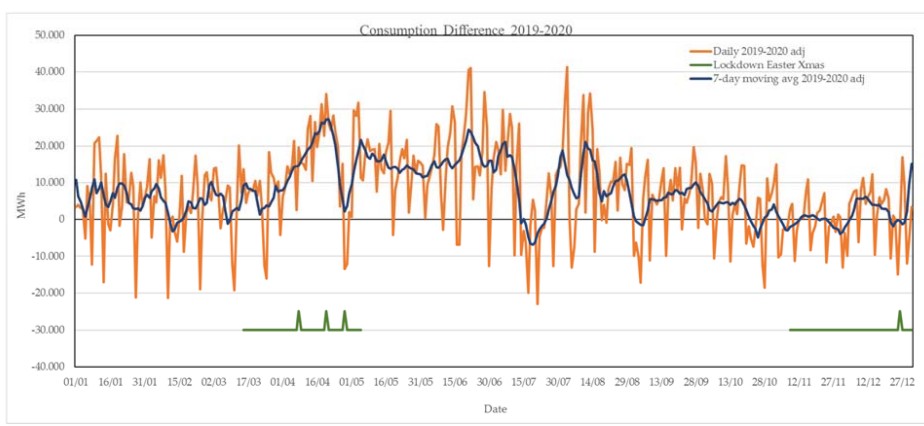

**Figure 4.** Reduction in daily electricity demand in the mainland grid (in MWh) in 2020, adjusted for temperature, from that og 2019 including 7-day moving average values (blue). The Easter holiday and lockdown periods are also indicated (Green).

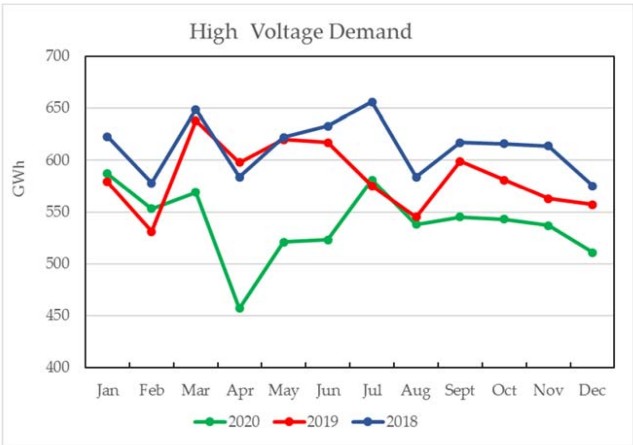

**Figure 5.** Monthly amounts of electricity demand of the high voltage customers for 2018, 2019, and 2020 (GWh).

Here the effect of the pandemic is much clearer. As travel embargoes were imposed both for domestic and international travel, electricity consumption started dropping in view of the strong dependence of the economy on tourism (GVA for the NACE sector of Wholesale and Retail Trade and Accommodation and Food Services for Crete making up 35% of total, with the Cyclades and Dodecanese islands reaching as high as 48% vs. 25% for Greece as a whole).

As seen in both Figure 6a,b, consumption in 2018 and 2019 is almost identical. In 2020, on the other hand, a clear deviation starts already in April and persists until September, with its maximum in June decreasing from July on as travel restrictions were gradually lifted. After the August tourism peak, the difference decreases to almost zero in November and remains so also in December. The pattern is the same in both Crete and the rest of the islands and follows tourist indices [4], such as international arrivals, where starting from March with the lockdown in effect for half of its days, the arrivals dropped to almost zero. The arrivals then started increasing but never reached more than 50% of those of 2019.

Focusing on the months of November and December, for which tourist arrivals in Crete and the Islands are very few, again the difference is almost zero which is in agreement with the minimal difference in the second lockdown, re-enforcing the conclusion—a relaxation of guidelines for teleworking notwithstanding—that the population has adapted its activity so as to circumvent the administrative restrictions.

The very low electricity demand starting in March significantly affected electricity prices driving them downwards. In Figure 7, the weekly average Marginal System Price (MSP) is shown for the years 2019 and 2020. Even though the MSP is a function of many variables including those of imports from neighboring countries, RES production, NG prices, and even ETS allowance prices on a longer timeframe, the sharp drop after week #12 (16–23 March) is primarily due to the lockdown demand decrease, although high generation of RES units and the decreased variable cost of the CCGT units because of the low prices of LNG in the LNG terminal of Revithoussa also contributed. It must be stressed that the average MSP of April 2020 is the lowest average MSP of the last four years, while also a number of hours of zero MSP occurred due to long periods of high wind generation. The difference in MCP between 2020 and 2019 is seen to follow that of both overall and high voltage industrial demand.

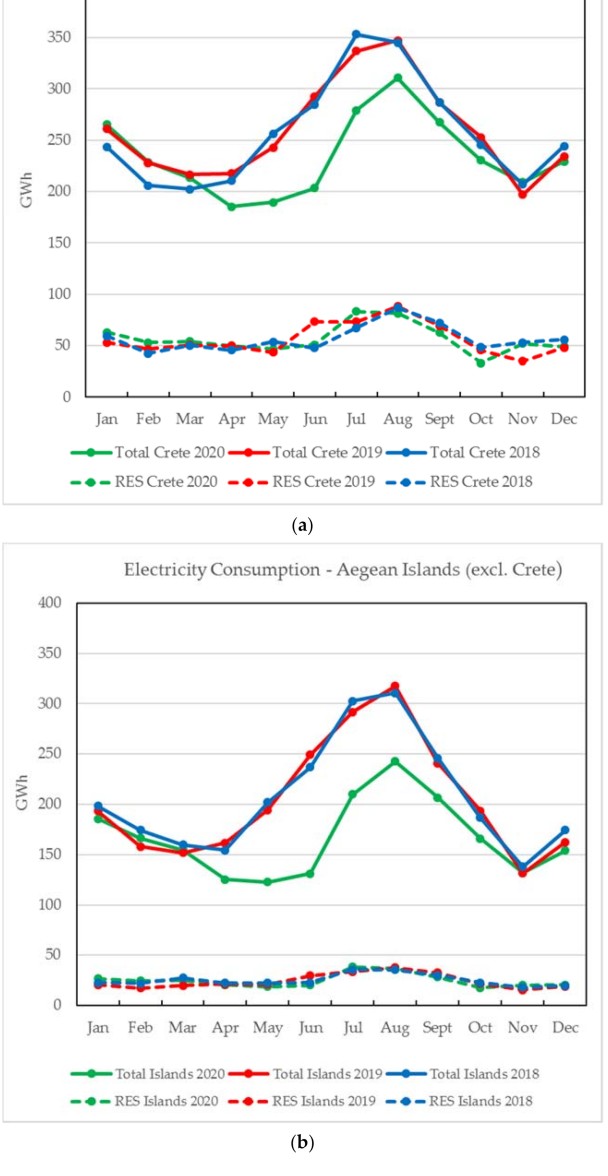

(**a**)

(**b**)

**Figure 6.** (**a**) Monthly amounts of total electricity consumption in Crete and generation from RES installations for 2018, 2019, and 2020 (GWh); (**b**) monthly amounts of total electricity generation by all stations in the Aegean islands (excluding Crete) and those from RES installations for 2018, 2019, and 2020 (GWh).

As a side effect, in week #19 (4–10 May 2020) and #21 (25–31 May), there was no lignite production notified in the day-ahead market of electricity, and on 8 June 2020 there was zero lignite production, a first since 1953 when the first lignite power plant went online in Greece. For the year, lignite production was down by 45% compared to 2019, reaching only 5713 GWh.

Natural gas consumption was also affected by the pandemic. The majority (more than 60%) of the NG volume is utilized for electricity production, yet the penetration of NG in the residential and tertiary sector is increasing as connections to the NG low pressure grid increase with an annual rate in the order of 4–6%. In Figure 8, the monthly consumption of NG for tertiary and residential use, excluding the amounts used for electricity production and heavy industry, with and without adjustment for temperature as described in the previous sector, is presented.

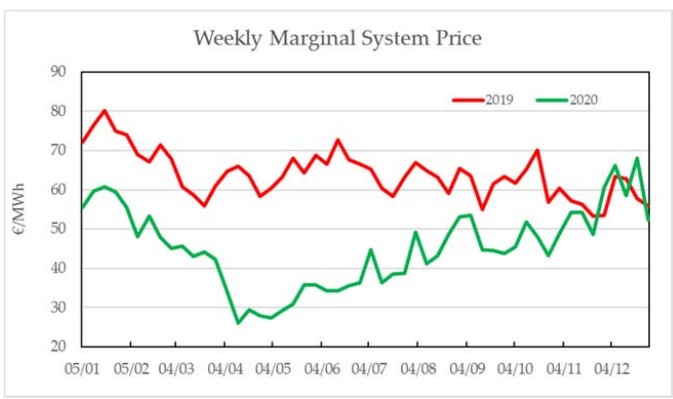

**Figure 7.** Weekly Marginal System Prices in 2019 and 2020 (in EUR/MWh).

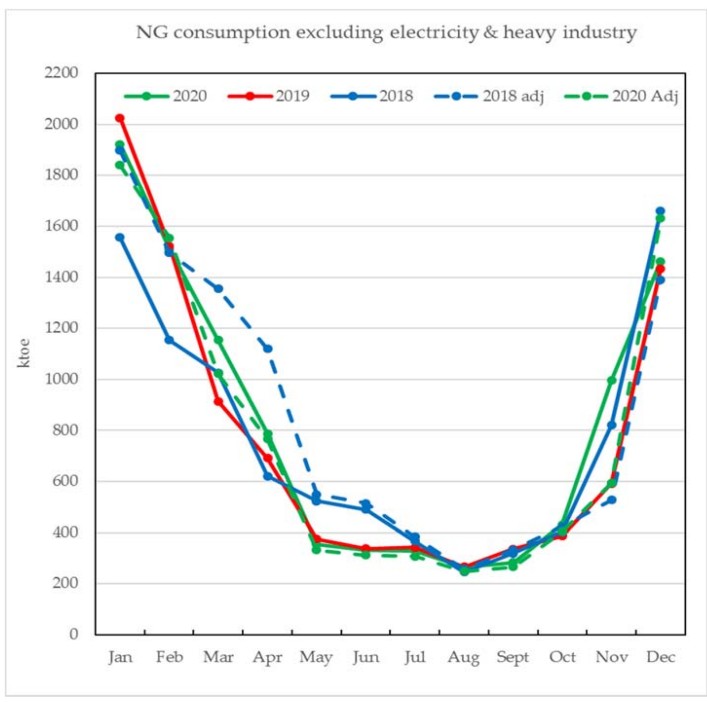

**Figure 8.** Monthly amounts of NG consumption excluding those for electricity production and heavy industry (in GWh).

Unlike electricity, consumption, both actual and adjusted, is higher in 2020 than in 2019 for the whole 10 months. The differences in NG consumption between 2019 and 2020

during the lockdown months of April, November, and December, although present, cannot be attributed with some degree of confidence to the measures and changes in behavior. The noticeable drop in May takes place after the lockdown was lifted on 4 May. The increase in actual consumption in November is most likely due to a temperature differential as evidenced by the much lower adjusted value.

Additionally, shown in Figure 9 is the consumption of diesel and LPG mostly for heating in the residential, tertiary, and light industries, both as reported and as adjusted for temperature effects which are pronounced in November to February of 2018–2019 in line with the HDD variation shown in Figure 2.

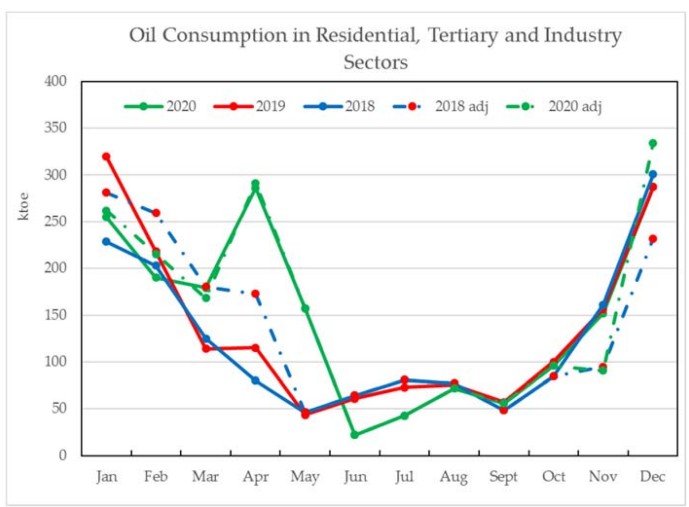

**Figure 9.** Comparison of heating oil consumption, actual and adjusted for temperature variation, for the years 2018, 2019, and 2020 (2020 provisional data).

Here the opposite pattern to that of the transport consumption is seen, with a large increase in the months of the first lockdown reaching over 100% in its peak in April. This, however, is partially due to the collapse of the oil prices (a drop of 15% in the transport Consumer Price Index between February and May which has not returned fully to the pre-COVID levels) in that period, with the retail price of heating oil reduced by almost 30% leading to off-season record sales, meant to be used in the next winter season. On a yearly basis, the increase in heating oil consumption, adjusted for temperature, in 2020 compared to 2019 was ca. 12%, that is, 165 ktoe, which is to be compared with a less than 1% difference between 2018 and 2019.

In Figure 10, the consumption of oil products in the inland transport sector is shown. As in the electricity consumption, a significant drop in the consumption of oil products in the transport sector is observed in the lockdown months of March–April with a gradual return to pre-COVID levels in May and June, after which time the difference decreased. Specifically, for the road transport, whereas the difference of consumption in the months of the lockdown between 2018 and 2019 is less than 3%, an overall 21% reduction was observed in 2020 for the month of March, during half of which the lockdown was in effect, and a and 37% reduction was observed in the month of April. This difference then decreased to 20% in May, 10% in June, and returned to pre-COVID levels in July. The imposition of the second lockdown on 7 November also had a noticeable effect on road gasoline and diesel consumption, although less pronounced than the spring one. The yearly reduction in all road transport fuels in 2020 compared to 2019 was ca. 13%, corresponding to 667 ktoe on the basis of preliminary data.

This pattern is confirmed by the data published (http://www.patt.gov.gr/site/index. php?option=com_content&view=article&id=37992, accessed on 3 January 2021) by the Region of Attica Traffic Control Center on 24 December 2020, which showed that the traffic in the Athens basin was 49.5% less in March 2020 compared to March of 2019, while for the

November–December second lockdown period the reduction was only 15.4% compared with the same period in 2019.

There is a significant effect in the internal navigation and domestic aviation sectors. Both have been affected by the lockdown prohibition of travel as flights and sailings have been reduced. Both of these means of transport are obliged to provide service despite reduced passenger and cargo loads, especially in the islands.

In Figure 11, the overall consumption of kerosene jet fuel per month for the three years 2018, 2019, and 2020 is shown. The effect of flight cancellations due to travel restrictions is clearly evident with the 2020 pattern having striking similarities with that of electricity consumption in the islands (Figure 6), reflecting again the influence of tourism. However, as no detailed data are currently available to desegregate 2020 domestic use, proxies have to be utilized. For domestic aviation, the ratio of the number of flights between 2020 and 2019 (58%) has been used to adjust 2019 consumption to reflect 2020 fuel (jet kerosene and gasoline) use.

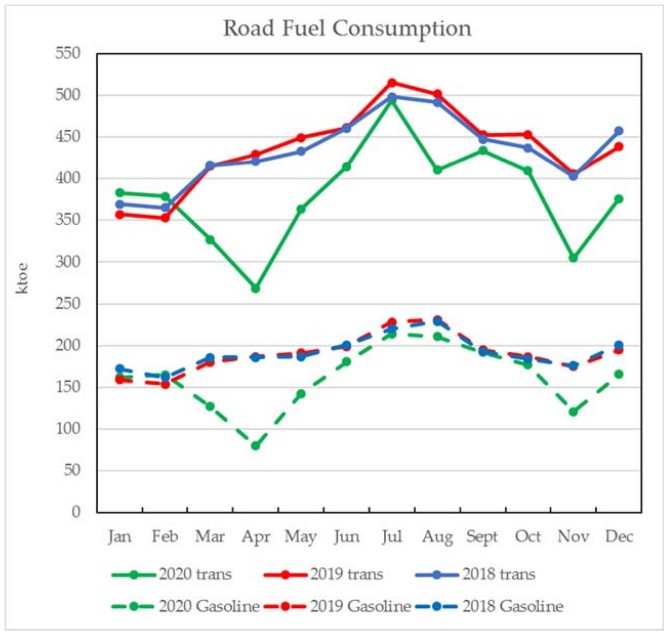

**Figure 10.** Comparison of road total fuel consumption and gasoline (in ktoe) for the years 2018 (blue), 2019 (red), and 2020 (green)—provisional data.

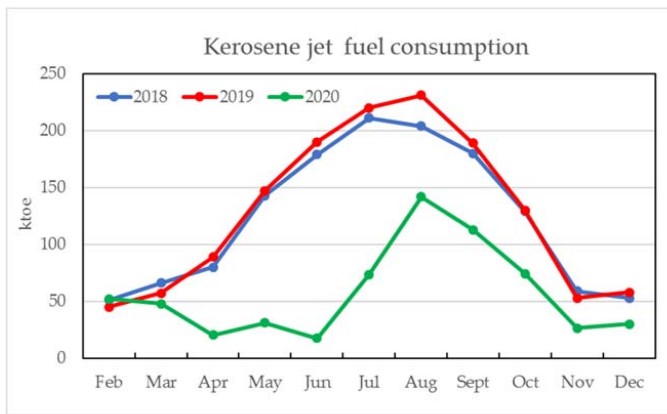

**Figure 11.** Inland consumption of kerosene jet fuel (in ktoe) for the years 2018, 2019, and 2020 (2020 provisional data).

For domestic navigation, the reduction in vehicles and passengers transported, i.e., 17.2% and 37.1%, respectively, in 2020 with regard to 2019 for the main harbor of Piraeus (http://www.olp.gr/el/investor-information/annual-reports, accessed on 1 March 2021), was taken as the proxy, with an overall reduction coefficient taken to be 25% in view of the fact that, especially in the lockdown months, commercial vehicles continued to be ferried at a higher rate than passengers, and the reduction in Piraeus harbor revenues was down 27%.

## 4. Impacts on GHG Emissions

### 4.1. Greek GHG Emissions Profile

GHG emissions in Greece starting from 1990, the United Nation Framework Convention for Climate Change (UNFCCC) base year, rose until 2005, a base year for a number of EU targets, and have been declining since. In Table 4, the evolution of the emissions taken from the latest submission (April 2020) of Greece to UNFCCC is provided.

**Table 4.** GHG emissions by sector (UNFCCC Inventory).

| (ktCO$_2$eq) | 1990 | 1995 | 2000 | 2005 | 2010 | 2011 | 2012 | 2013 | 2014 | 2015 | 2016 | 2017 | 2018 | 2019 |
|---|---|---|---|---|---|---|---|---|---|---|---|---|---|---|
| **Energy** | 77,026 | 81,091 | 96,797 | 107,297 | 93,155 | 92,036 | 88,304 | 77,926 | 74,491 | 71,190 | 66,966 | 70,259 | 67,307 | 61,224 |
| **IPPU** | 11,277 | 13,603 | 15,193 | 15,432 | 11,760 | 10,424 | 11,245 | 11,966 | 12,328 | 11,995 | 12,503 | 12,794 | 12,387 | 11,668 |
| **Agriculture** | 10,140 | 9488 | 9147 | 8959 | 8839 | 8596 | 8405 | 8405 | 7990 | 7846 | 7856 | 7888 | 7781 | 7714 |
| **Waste** | 4864 | 5,151 | 5356 | 4758 | 4769 | 4537 | 4409 | 4409 | 4468 | 4451 | 4515 | 4646 | 4746 | 5001 |
| **Total** | 103,309 | 109,332 | 126,492 | 136,446 | 118,522 | 115,593 | 112,323 | 102,7( | 99,277 | 95,48; | 91,840 | 95,58( | 92,222 | 85,607 |
| **LULUCF** | −2107 | −2872 | −1941 | −3301 | −3043 | −3131 | −3086 | −1582 | −126 | −3719 | −3473 | −3209 | −2978 | −3300 |

NB1: IPPU comprises industrial processes and products use including use of PFC, HFC and SF6. NB2: 2019 data provisional.

Total emissions are seen to increase gradually from 103.3 MtCO$_2$eq in 1990 to a maximum of 136.5 MtCO$_2$eq in 2005 and to decrease to a minimum of 91.9 MtCO$_2$eq in 2016, at which general level they remain until 2018.

The energy sector, which in UNFCCC Inventories includes energy related emissions of ttransport as well as of the industry and residential/tertiary sectors, accounts for close to three quarters of the emissions with its share ranging from ca. 75% in the 1990–2000 decade up to ca. 80% in the 2004–2013 period, with a gradual return to ca. 73–74% in the last five years. Of the other three sectors in the Inventory, waste emissions have remained virtually unchanged at about 4.7 MtCO$_2$eq, as have those of the agriculture sector at about 7.7 to 7.9 MtCO$_2$eq. Finally, the Industrial Processes and Product Use (IPPU) sector exhibits a trend similar to the overall emissions increasing from about 11 MtCO$_2$eq in 1990 to a maximum of 16.4 MtCO$_2$eq by 1999 and declining back to 11–12 MtCO$_2$eq in the last ten years until 2019. It is thus worth focusing on the energy sector emissions in view of their major contribution and their volatility.

In the UFCCC inventory energy sector, the energy industries subsector (electricity generation and refineries) which comprises mostly the electricity generation activity is seen (see Table 5) to account for more than 50% in the 2000–2010 period, down to 39% in 2018 and 32% in 2019, of total energy sector emissions. The reduction energy industries emissions from 55.01 MtCO$_2$eq in 2010 to 31.97 MtCO$_2$eq in 2019 (to which one should add an additional ca. 0.9 MtCO$_2$eq coming from mostly CH$_4$ fugitive emissions at the lignite mines) is almost entirely due to the electricity emissions reduction as their percentage varied only between 76% in 2020 and 84% in 2019.

A more detailed look at electricity sector emissions is provided by the latest (April 2020) EU ETS verified emissions presented in Table 6, in which the 2020 values have been estimated using the latest available (2019) emission factors.

After the contraction in electricity emissions due to the 2010–2014 crisis, the precipitous drop in 2016, which was the year with the lowest GNP of the last 10 years, is due also to the

almost doubling of the electricity produced by NG that was caused by the very large drop in NG prices, accompanied with a similar decrease in lignite production. As the economy started recovering starting in 2017, in the second half of 2018 the ETS allowance prices started increasing from ca. EUR 5/EUA in 2016–2017 to EUR 8/EUA in the end of 2018 and continued to climb to ca. EUR 22/EUA in 2019 reaching EUR 28–30 in 2020 with a sharp drop in April/May down to ca. EUR 16/EUA from ca. EUR 25/EUA in January/February 2020, and, despite another smaller downward spike in November, on to ca. EUR 30/EUA by the end of 2020. The negative spikes seen in 2020 correspond to the first and second waves of the pandemic, which were of limited extent and with quick recovery.

**Table 5.** Energy Sector (UNFCCC inventory) Emissions.

| CO$_2$eq (Mton) | 2010 | 2011 | 2012 | 2013 | 2014 | 2015 | 2016 | 2017 | 2018 | 2019 |
|---|---|---|---|---|---|---|---|---|---|---|
| Energy Industries | 55.01 | 55.76 | 50.28 | 46.79 | 41.70 | 41.70 | 37.57 | 40.58 | 38.89 | 31.97 |
| of which Electricity | 42.23 | 44.78 | 46.53 | 41.69 | 38.80 | 36.57 | 32.22 | 35.81 | 33.53 | 27.32 |
| Industry | 4.98 | 5.52 | 5.28 | 5.46 | 5.24 | 5.24 | 5.35 | 5.78 | 5.11 | 4.62 |
| Transport | 20.06 | 16.69 | 16.52 | 16.50 | 17.05 | 17.05 | 17.38 | 17.18 | 17.40 | 17.81 |
| Other Sectors | 11.21 | 19.58 | 5.17 | 5.08 | 6.49 | 6.63 | 6.13 | 6.13 | 5.43 | 5.93 |
| Others | 1.24 | 1.31 | 1.18 | 1.13 | 1.02 | 1.02 | 0.77 | 0.84 | 0.76 | 0.89 |
| **Total** | **92.50** | **98.86** | **78.44** | **74.96** | **71.49** | **71.63** | **67.20** | **70.51** | **67.59** | **61.22** |

**Table 6.** Electricity Verified Emissions.

| (ktCO$_{2eq}$) | 2010 | 2015 | 2016 | 2017 | 2018 | 2019 | 2020 (Estimated) |
|---|---|---|---|---|---|---|---|
| Lignite | 35,615 | 29,408 | 23,127 | 25,592 | 24,141 | 17,459 | 9574 |
| NG | 2949 | 3684 | 5586 | 6553 | 6013 | 6721 | 7536 |
| Oil | 3661 | 3485 | 3503 | 3661 | 3373 | 3431 | 2864 |
| Electricity | 42,225 | 36,573 | 32,216 | 35,806 | 33,528 | 27,610 | 19,974 |
| Total—ETS | 54,924 | 49,886 | 46,310 | 49,262 | 47,105 | 40,476 | N/A |
| Total—National | 118,436 | 953,303 | 91,698 | 95,421 | 92,222 | 85,607 | N/A |

At the same time, NG prices also went from ca. EUR 25/MWh in 2017 and 2018 to over EUR 30/MWh in 2019 which was followed by a large drop down to ca. EUR 22/MWh in all 2020. The high EUA price led to a 25% drop of lignite production in 2019, which, combined with even higher EUA prices in 2020 and NG prices dropping by 25% to ca. EUR 22/MWh, led to halving of the emissions from lignite generation in 2020 and a further increase in NG generation. It should be also mentioned that the emissions from oil powered plants which cover exclusively the demand in the Aegean islands remain virtually constant from 2010 to 2019 but drop by ca. 17% in 2020.

Whereas the electricity sector emissions dropped by over ca. 35% in the last 10 years, the second larger contributor to emissions, the transport sector, has decreased much less (ca. 15%). The major reduction has taken place in the 2011–2012 period of the crest of the economic crisis and the emissions have remained at the same level of about 17 MtCO$_2$eq as the renewal of the fleet with less polluting vehicles remained very small.

A similarly large decrease in emissions in 2012–2013 is also seen in the Other Sectors category which includes direct energy use in the residential and tertiary and agriculture

sectors, again reflecting the economic stress especially in households which resulted in a reduction of energy (mainly for heating) expenditure.

*4.2. COVID-19 Effect on Emissions*

The effect of lockdown reduces economic activity and affects the behavior of the population as mobility is constrained by decree or by choice. In this analysis, as discussed in the previous Section 3, the focus is on the electricity, heating, and transport sectors. These sectors accounted for over 55% of the national emissions in 2019.

In the electricity sector, as discussed in Section 3, there is a clear reduction in demand in 2020 of ca. 3.48 TWh, which is mostly seen during the first lockdown in March–April, although it persisted to a lesser amount for the rest of the year. How this translates into emissions reduction, however, is not clear as it depends on the generation mix and net import amounts which are affected only tangentially by the lockdown through fuel and EUA prices. In Table 6, the estimated 2020 emissions from electricity production are presented, which were computed from the actual production with the use of the 2019 emission factors for lignite, NG, and oil (1.675 $tCO_2eq$/MWh, 0.414 $tCO_2eq$/MWh, and 0.748 $tCO_2eq$/MWh, respectively).

If the difference in electricity consumption between 2019 and 2020 (corrected for temperature) of 2.76 TWh is taken as a measure of the COVID-19 impact on mainland electricity emissions and the same ratio of lignite and NG production to total consumption per month is utilized, the resulting drop in emissions is estimated at 0.771 $MtCO_2eq$. A fairer estimate of the effect on emissions should instead be made assuming the percentages of lignite and NG to total generation of 2019 are applicable also in 2020 to account for the fact that the lignite production in 2020 was halved, not because of the pandemic, but because of the high cost of EUAs and the low cost of NG. This would lead to total emissions reductions in the mainland grid of 1243 $MtCO_2eq$.

In the Aegean islands where there are no electricity imports and all emissions come from oil combustion, as seen in Figure 6a,b, and the production from RES is more or less constant, the difference in consumption would lead to a reduction in oil use of 764 GWh, which corresponds to 0.571 $MtCO_2eq$.

As a result, the total impact of the pandemic is estimated at 1.341 $MtCO_2eq$ (or 1.814 $MtCO_2eq$ with analogous lignite emission contribution). Of that amount, ca. 1 $MtCO_2eq$ can be assigned to reduced tourist activity (double that of the emissions attributed to the islands in view of the equal parts of arrivals reduction in the mainland and the islands—see [4]).

The lockdown also affected the transportation GHG emissions, as well as other pollutants [10]. In fact, the ground transportation sector's emissions can be very responsive to policy changes and economic shifts. Ground transport accounts for nearly half the decrease in emissions during confinement. A concurrent increase in active travel (walking and cycling, including e-bikes) might help if it is maintained to cut back $CO_2$ emissions and air pollution as confinement is eased [19].

In Table 7, the effect of the reduction in road mobility GHG emissions is clearly evidenced. The difference between the emissions in 2020 and 2019 is on the order of 12% as opposed to 1% between 2018 and 2019.

**Table 7.** Road transport emissions (kt$CO_2$). Lockdown periods shaded.

|      | Jan  | Feb  | Mar  | Apr  | May  | Jun  | Jul  | Aug  | Sept | Oct  | Nov  | Dec  | Year   | %    |
|------|------|------|------|------|------|------|------|------|------|------|------|------|--------|------|
| **2020** | 1121 | 1109 | 957  | 787  | 1066 | 1216 | 1449 | 1202 | 1272 | 1199 | 893  | 1100 | 13,370 | 0.87 |
| **2019** | 1045 | 1034 | 1216 | 1257 | 1317 | 1352 | 1510 | 1471 | 1327 | 1327 | 1188 | 1285 | 15,327 | 1.00 |
| **2018** | 1082 | 1070 | 1219 | 1234 | 1269 | 1351 | 1461 | 1442 | 1312 | 1281 | 1181 | 1339 | 15,241 | 0.99 |

This difference is very pronounced in the months of April and November–December, during which the tourist activity is very small and evidently is an indication of the behavior

of the native population. There is also a reduction in August, smaller than those during the two lockdown periods, which most likely is due to the reduction of tourist activity. The yearly road reduction is 1957 MtonCO$_2$eq, which is ca. 2% of total national emissions.

The other transport means, rail, domestic navigation, and aviation, have contributed less than 15% of the total transport emissions in the last years, with navigation, domestic aviation, and rail contributing ca. 11%, 3% and less than 1%, respectively. As no detailed data are available yet for domestic aviation consumption from September to December 2020, taking the reduction in the number of domestic flights (42%) as a proxy, the emissions in 2020 can be estimated to be 0.173 MtonCO$_2$eq (down from 0.410 MtonCO$_2$eq in 2019 provisional Inventory data). Similarly, for internal navigation, in view of the fact that connections to the islands have schedules that are not directly dependent on the number of passengers (who are down ca. 35% in the first nine months of 2020 compared to either 2019 or 2018) and extrapolating available data from January to July 2020, a similar percentage of 2019 emissions (2068 MtonCO$_2$eq provisional inventory data) reduction of 25% would lead to ca. 0.54 MtonCO$_2$eq fewer emissions.

This would result in an overall reduction of emissions in the transport sector of ca. 2.688 MtonCO$_2$eq.

The effect on emissions from the use of heating oil and NG in the residential and tertiary sectors is shown in Table 8. For both oil and NG use, an increase is seen, after adjustment for temperature, on the order of 10% in 2020 compared to both 2019 and 2018, with the total amount estimated at 0.544 MtCO$_2$eq.

**Table 8.** Emissions from heating oil and NG in residential, tertiary, and industrial sectors (ktCO$_2$). Lockdown periods shaded.

| Oil | Jan | Feb | Mar | Apr | May | Jun | Jul | Aug | Sept | Oct | Nov | Dec | Year | % |
|---|---|---|---|---|---|---|---|---|---|---|---|---|---|---|
| **2020** | 738 | 547 | 515 | 830 | 451 | 53 | 115 | 200 | 153 | 270 | 435 | 833 | 5140 | 1.11 |
| **2020 Adj** | 759 | 622 | 483 | 845 | 451 | 53 | 115 | 200 | 153 | 270 | 255 | 972 | 5179 | 1.12 |
| **2019** | 929 | 629 | 323 | 327 | 117 | 167 | 203 | 210 | 156 | 282 | 448 | 833 | 4624 | 1.00 |
| **2018** | 660 | 585 | 356 | 224 | 125 | 178 | 226 | 215 | 131 | 238 | 462 | 873 | 4272 | 0.92 |
| **2018 Adj** | 815 | 751 | 519 | 497 | 125 | 178 | 226 | 215 | 131 | 238 | 268 | 670 | 4632 | 1.00 |
| **NG** | **Jan** | **Feb** | **Mar** | **Apr** | **May** | **Jun** | **Jul** | **Aug** | **Sept** | **Oct** | **Nov** | **Dec** | **Year** | **%** |
| **2020** | 385 | 303 | 231 | 158 | 71 | 67 | 66 | 53 | 57 | 86 | 200 | 293 | 1971 | 1.07 |
| **2020 Adj** | 369 | 312 | 205 | 154 | 67 | 63 | 62 | 50 | 53 | 81 | 119 | 327 | 1860 | 1.01 |
| **2019** | 406 | 305 | 183 | 139 | 75 | 68 | 69 | 54 | 67 | 78 | 119 | 288 | 1850 | 1.00 |
| **2018** | NA | NA | NA | NA | NA | NA | NA | 49 | 64 | 81 | 165 | 333 | 691 | 1.14 |
| **2018 Adj** | NA | NA | NA | NA | NA | NA | NA | 52 | 67 | 85 | 104 | 277 | 584 | 0.97 |

In the IPPU UNFCCC Inventory category, more than half of GHG emissions come from the contribution of F-gases which are not expected to be affected noticeably by the lockdown. The rest, mostly process emissions in the cement and lime industries which supply the construction sector, are expected to be affected by the downturn of the overall economic activity and, in particular, of tourism whose infrastructure upgrading and enlarging comprised a large percentage of pre-COVID construction, but no data are available at this time for an estimate.

Looking at the other main UNFCCC inventory categories, no reduction should be expected in the Agriculture and Waste ones as these activities are not directly affected by the lockdown.

## 5. Discussion and Conclusions

The results on energy and GHG emissions presented in the previous sections are summarized in Table 9. Overall, a noticeable reduction in emissions should be expected in 2020 which is estimated at 3.48 $MtonCO_2eq$ or 3.95 $MtonCO_2eq$ if the lignite electricity had remained at the same percentage of total production of the previous years.

**Table 9.** Changes 2020–2019 due to COVID impacts.

| Sector | Energy (GWh) | Emissions (MtCO$_2$) | Emissions (% Wrt 2019) |
|---|---|---|---|
| **Electricity** * | −3528 | −1.34 | −4.9% |
| Mainland grid | −2764 | −0.77 | −3.2% |
| Islands and Crete | −764 | −0.57 | −16.6% |
| **Transport** | −10,286 | −2.69 | −15.1% |
| Road | −7797 | −1.97 | −12.8% |
| Navigation | −1818 | −0.54 | −26.3% |
| Aviation | −671 | −0.17 | −42.0% |
| **Heating and other use** * | 1916 | 0.54 | 8.4% |
| **Total** | −11,898 | −3.48 | −6.7% |
| **Total with lignite adjust.** | −12,263 | −3.95 | −7.6% |

* Temperature adjusted.

The results presented in the previous Sections 3 and 4 lead to two main conclusions. The first refers to the magnitude of the energy use and emissions reductionm and the second to the noticeable difference between the two lockdown periods. Even though all countries have experienced two waves (so far) of the pandemic, the timing of the crests and ebbs is different, and the national responses also differ both as to specific measures taken including the different activities closed by decree, as well as to their duration and rate of the subsequent opening by activity. The structure and strength of each economy also differs from country to country. It is thus important to state that the results presented here and the conclusions drawn from them refer to Greece and may not apply directly to other countries.

Starting with the impact of the COVID-19 measures on GHG emissions, one should note that the total reduction is the result of a decrease in the electricity consumption and the road and navigation transport sector activity and an increase in the use of diesel oil and NG mostly for heating and for other uses in the residential and tertiary sectors. The electricity consumption is lower in the residential and tertiary sectors despite the increase in the use of electric appliances for working at home, accompanied by an even deeper percentage decrease in electricity use in the industrial sector.

The travel restrictions and their effect on tourism are seen to account for a third of the emission reduction and are highlighted in the electricity consumption in Crete and the Aegean Islands where most of the reduction is found in the tourist period of May to October.

The recovery after the first lockdown was lifted is seen to be, unlike its initiation, gradual. This is to be attributed to the timing of the first lockdown whose lifting coincided

with the beginning of the tourist season that itself was severely affected, and although it slowly started again, it never fully returned to the levels of previous years.

The nature of the second lockdown in November is different and worth analyzing as for these two months at the end of the year the incoming tourism is minimal, and so is its effect. In this second lockdown, the electricity consumption in the mainland grid, adjusted for temperature, is seen to be on the same order as those of the previous years. This is more clearly the case in Crete and the rest of the Aegean islands. As tourist activity is negligible in these winter months, the implication is that the general public has adjusted to or partially ignored lockdown measures and returned to pre-COVID behavior. This is not the case for oil consumption in road transport, which is due in part to the restriction of outdoor circulation after 2100 h. The drop, however, is smaller than that of the first lockdown partly due to a decrease in Government teleworking guidelines from 80% in the first lockdown to 50% in the second and accommodation by the public, but also to circumvention of the measures.

The overall tentative conclusion is that as the initial strong fear of mortality in the initial stages subsided, the public has found ways to align its behavior with the measures so as to return as much as possible to previous patterns, as expected [20]. This is may also reflect the reduced efficacy of the measures due to faulty design or to inadequate policing. In short, the reduction in GHG emissions seen in 2020 caused by measures addressing the pandemic, at least in Greece, is not expected to continue in the coming years as no indications of permanent changes in behavior are discernable despite the increase in teleworking.

This conclusion is further supported by preliminary data from the extension of the lockdown in the first part of January 2021, which was lifted gradually after the first week of the year with only primary schools opening on 11 January and retail stores opening, with restrictions, on 18 January 2021. According to the Traffic Control Department of the Region of Attica (http://www.patt.gov.gr/site/index.php?option=com_content&view=article&id=38338&catid=3&Itemid=709, accessed on 3 February 2021), traffic actually increased in January 2021 by 1–5% compared to the same period in 2020. A possible explanation for this increase could be the reluctance of the public to use the means of mass transit (drop of ridership by 50% following the lockdown in November (https://data.gov.gr/datasets/oasa_ridership/, accessed on 19 February 2021) to avoid congestion for fear of contagion and turning instead to private vehicle use. This tentative conclusion that the overall residual effect of COVID measures will be small remains to be confirmed when information from other sectors of activity and covering longer periods as measures are further relaxed becomes available.

In the short term (ca. 2021–2022), a reduction in GHG emissions in Greece from the lingering effects of COVID-19 should again be expected, mostly from reduced tourism activity in 2021 in view of the still real concern for infection, which, however, will be much less than in 2020 as a larger part of the population is vaccinated. Beyond this immediate period, a more permanent decrease may be caused by a change in mobility patterns if on-line work and the use of soft transportation means, hopefully, an increase.

In the case of some patterns of behavior, especially regarding energy use in the residential sector and transport activity seen in the lockdown periods to take a more permanent form, their effect would need to be included in the revision of the Greek National Energy and Climate Plan to meet the new target of a 55% reduction in GHG emissions in the EU by 2030 as set in December 2020.

**Author Contributions:** D.L. conceived the paper scope and the theoretical framework, analyzed the results and co-wrote the paper. N.G., E.G., S.M., Y.S., and H.D. analyzed the results and designed the results and co-wrote the paper. All authors have read and agreed to the published version of the manuscript.

**Funding:** This research was funded by European Commission Horizon 2020 Framework Programme, "PARIS REINFORCE" Research and Innovation Project, grant number 820846. The APC was funded by the National Technical University of Athens.

**Institutional Review Board Statement:** Not applicable.

**Informed Consent Statement:** Not applicable.

**Data Availability Statement:** Publicly available datasets were analyzed in this study.

**Acknowledgments:** This work has been carried out in the scope of Contract DG-2005-60591 with the European Climate Foundation whose financial support is gratefully acknowledged. It is also based on the H2020 European Commission Project "PARIS REINFORCE" under grant agreement No. 820846. The sole responsibility for the content of this paper lies with the authors. The paper does not necessarily reflect the opinion of the European Commission.

**Conflicts of Interest:** The authors declare no conflict of interest.

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
