# Peer review of "Energy and GHG Emissions Aspects of the COVID Impact in Greece"

_energies, doi:10.3390/en14071955_

Round 1

Reviewer 1 Report

I consider that the article will benefit if the author makes remarks within the manuscript regarding the following aspects:

  1. What is the main contribution of the article? Please explain in the introduction.
  2. What is the overall structure of the article? Please explain in the introduction.
  3. The analysis of the existing research literature is not very clear, please refine it.
  4. All graphics are not very clear to see, please make corrections.
  5. The structure of section 2 is a bit confusing, please make it more clear. For example, separate Materials and Methods.

Reviewer 2 Report

Dear Authors

A very interesting and extensive article. The amount of collected and processed data is impressive. Correct methodology. I consider the applied simplifications related to certain data's current unavailability (e.g., energy balance 2020) as acceptable.

Substantive comments.

Line 295 - "is the very high correlation of the changes" - what that means. Has the correlation been calculated?

The description of Fig. 4 (lines 304-311) refers to data from 2018. I didn't find it on the chart. Are they elsewhere?

Discussion and conclusions correct, although not fully supported by the conducted research. Statement: "A partial cause of this increase is the reluctance of the public to use the means of mass transit to avoid congestion for fear of contagion and turning instead to private vehicle use." it would additionally require sociological research. Limiting the use of mass transport by society may also be caused by remote work, unemployment, or other factors resulting from the pandemic.

Editorship Notes.

All plots should have annotated axes. I would replace the numbering of days with months, which would improve the readability of the charts. The specific dates can be entered in the description.

Fig. 3 is illegible. Maybe starting at 80,000 to 200,000 it will be easier to analyze dependencies.

The same notation of numbers should be used throughout the article:

 0.771MtCO2eq - without spaces or 1.243 MtCO2eq - with spaces.
